# OpenReview forum: "Self-Rewarding Rubric-Based Reinforcement Learning for Open-Ended Reasoning"
_ICLR.cc/2026/Conference — Submitted to ICLR 2026_

### Official Review · Reviewer_LEaX · 2025-10-31

**Soundness:** 2
**Presentation:** 2
**Contribution:** 2
**Rating:** 2
**Confidence:** 4

**Summary:**

This paper introduces self-rewarding rubric-based RL for open-ended reasoning task, i.e., HealthBench. Firstly, the paper conducts a meta-evaluation on open-source models, and qualitatively evaluates the ability of Qwen3-32B and Qwen3-8B on meta-evaluation. Then, it introduces a self-rewarding RL framework for improving training efficiency and open-ended downstream tasks.

**Strengths:**

- This paper is well-written and structured.
- Introduce a self-rewarding RL framework for open-ended downstream tasks.

**Weaknesses:**

I am concerned that the novelty and soundness of this method are limited. On the one hand, previous work already introduced similar ideas, such as [1], [2]. Only the training objective introduces variance. On the other hand, this paper only verifies the effectiveness on HealthBench. I believe that introducing more benchmarks, such as creating writing, is more convincing.

[1]Constitutional AI: Harmlessness from AI Feedback;
[2]Self-Rewarding Language Models;

**Questions:**

Same to Weaknesses.

---

> ### Author Response · Authors · 2025-11-19
>
> Thank you for the thoughtful comments. We have already cited both works and discussed the differences in the paper: Constitutional AI focuses primarily on aligning with human preferences, while Self-Rewarding Language Models trains instruction following via DPO. As several recent papers have pointed out (e.g., [arXiv:2506.16507]), such pairwise preference–based training is prone to reward hacking, making it difficult to learn the truly desirable properties of responses. In contrast, our method differs both in training paradigm (RLVR with *task-specific rubrics*) and in target task (*open-ended reasoning*).
>
> Let me further clarify the novelty and motivation. To mitigate reward hacking, genRM-based scoring is becoming increasingly popular. However, genRM introduces substantial computational overhead, especially in open-ended tasks where the evaluation criteria vary widely across prompts. Our approach leverages self-rewarding, enabling almost **no idle GPU time during synchronous training**. This effectively scales reward computation together with RL computation, allowing a relatively small model to reach state-of-the-art performance on a single complex task when using less than half of the GPU hours in a run for baseline.
>
> In addition, we conducted experiments on Creative Writing V3, which further demonstrates the method’s generality beyond the medical domain. In this setting, we follow the same principle: each response is evaluated using single rubric out of total 22 rubrics, allowing the model to solve a simpler and more verifiable scoring problem (an easier task, a more accurate reward) before aggregating rubric scores.
>
> We trained Qwen3-8B with a configuration similar to the main experiments for 10 epochs (310 steps), and used claude-sonnet-4-20250514 for grading. The results are shown below:
>
> | model        | grading_temp | step | score |
> |--------------|--------------|------|-------|
> | Qwen3-8B     | -            |  -   | 51.83 |
> | Qwen3-8B-SR  | 0.6          | 155  | 61.82 |
> | Qwen3-8B-SR  | 0.6          | 310  | 61.05 |
> | Qwen3-8B-SR  | 1.0          | 155  | 63.22 |
> | Qwen3-8B-SR  | 1.0          | 310  | 59.40 |
>
> We observe consistent improvements across different grading temperatures just like the experiments in HealthBench. We hope these  address your concern.

---

### Official Review · Reviewer_GKXq · 2025-10-31

**Soundness:** 3
**Presentation:** 2
**Contribution:** 2
**Rating:** 4
**Confidence:** 3

**Summary:**

This paper focuses on the research question of "can the model grade itself using those rubrics and get better at both reasoning and grading?" They adapt the GRPO/DAPO-style RL and replace the external reward model with the policy itself, scoring each rubric item generatively and summing normalized positive rubric points as the reward (without a KL penalty). They train Qwen3‑32B on 4 K HealthBench-Easy prompts, achieving self-rewarding yields of HealthBench-Hard (graded by GPT-4.1), outperforming their RL baseline and a reported GPT-5 score.

**Strengths:**

1. The self-rewarding method sounds novel. The policy model doubles as a generative rubric scorer, yielding fine‑grained, verifiable eward signals without a separate reward model.  It is a crisp, impactful tweak that measurably improves performance and efficiency on a hard, open‑ended benchmark.
2. This method also sounds promising but would benefit from frozen‑grader/KL/length‑control ablations and multi‑grader/human eval to preempt known pitfalls (grader drift, reward hacking, verbosity bias).
3. The baseline comparison is complete. This paper first runs a “standard” rubric‑graded RL baseline, then switches the grader to the policy and compares.
4. This paper explains why self‑rewarding is faster in their infra and quantifies the effect

**Weaknesses:**

1. **This is the major concern of this paper**. All results are in the medical domain and on a single benchmark (HealthBench). Generalization to other open‑ended tasks is untested and explicitly left for future work.
2. The paper cites a GPT‑5 score (Table 4). Because this is a single benchmark + single grader (GPT‑4.1), the comparison should be framed cautiously and replicated under multiple graders/humans.
3. The paper omits KL divergence but does not give a reason.
4. The experimental study is not comprehensive. I do acknowledge that this paper tries many open-source models, but most of them are from the Qwen & Deepseek series. Including more diverse models, such as llama, gemma, will make the evaluation more convincing.

**Questions:**

Questions:
1. Why omit KL divergence?
2. How robust is self‑grading? If freezing the grader to the initial policy (vs moving with training), how do scores and stability change?
3. Any evidence that the policy learns to parrot the rubric triggers? That's to say, any reward hacking checks?
4. How does the approach perform on non‑medical, open‑ended datasets using different rubric styles?

Suggestions:
1. Add at least one non‑medical, open‑ended task with rubrics (e.g., instruction‑following or advice) to test portability of “policy‑as‑grader.” Even a small‑scale run would reduce the domain‑restriction limitation.
2. This method removes KL and uses the moving policy as a grader. Providing a 2×2 ablation: {frozen GRM, moving GRM} × {no‑KL, small‑KL} will be better.
3. The citation seems to have a few issues. Specifically, in line 644, there is something called "5 Team", but I have no idea where it comes from. Also, I am not sure why some citations have TLDR, but some do not. (This is a minor question raised during the review of the whole paper, which does not impact my decision of ratings).

---

> ### Author Response · Authors · 2025-11-19
>
> Thank you very much for the helpful and detailed comments.
>
> *Response to Weakness*
>
> **Results outside the health domain**
>
> We chose HealthBench Hard because it provides a rubric-based evaluation setting that is sufficiently challenging (even frontier models score below 50% accuracy). As a complement, we also conducted experiments on the Creative Writing V3 task from EQ-Bench, which measures a model’s creative-writing capability.
>
> Specifically, we used 1,000 synthetic creative-writing prompts and applied 22 rubrics (e.g., Emotionally Engaging, Weak Dialogue) that allow the model to self-evaluate during training. The scoring strategy matches the one used in the paper: the model evaluates one rubric at a time and aggregates the scores at the end.
>
> We trained Qwen3-8B with a configuration similar to the main experiments for 10 epochs (310 steps), and used claude-sonnet-4-20250514 for grading. The results are shown below:
>
> | model        | grading_temp | step | score |
> |--------------|--------------|------|-------|
> | Qwen3-8B     | -            |  -   | 51.83 |
> | Qwen3-8B-SR  | 0.6          | 155  | 61.82 |
> | Qwen3-8B-SR  | 0.6          | 310  | 61.05 |
> | Qwen3-8B-SR  | 1.0          | 155  | 63.22 |
> | Qwen3-8B-SR  | 1.0          | 310  | 59.40 |
>
> As shown, we observe consistent improvements across different grading temperatures, suggesting that our method generalizes beyond the medical domain.
>
> **Other evaluator choice**
>
> We have added the Creative Writing results as complementary evidence. For HealthBench, we also evaluated using Qwen3-32B and Kimi-K2, as shown in Figures 4 and 6(b).
>
> However, using GPT-4.1 as the 'oracle' grader is a reasonable choice: it achieves the highest HealthBench Meta score among available models, and the HealthBench paper reports that it surpasses the average physician performance in 5 out of 7 themes. Therefore, although multi-grader or human replication is valuable, GPT-4.1 provides a strong and well-validated reference point for evaluation.
>
> **Ablation on KL**
>
> We follow the recipe introduced in DAPO, which removes the KL penalty to **encourage exploration in reasoning-style tasks**. To isolate the effect of KL, we conducted an ablation on Qwen3-8B (note that we resolved the training-instability issue on Qwen3-8B by adding repetition penalty during scoring sampling). The HealthBench Hard results are as follows:
>
> - Qwen3-8B: 0.108
> - Qwen3-8B SR (without KL): **0.363**
> - Qwen3-8B SR (with KL): 0.289
>
> These results show that adding the KL penalty is indeed harmful for this task, confirming that removing KL leads to better performance in our setting.
>
> **More models**
>
> Thanks for your advice. Due to time limitation, we add the HealthBench Meta score for Llama-3.1-8B-Instruct(0.523) and gemma-3-27b-it(0.604) here, hoping it would be helpful for the community.

---

> > ### Author Response · Authors · 2025-11-19
> > **(Continued)**
> >
> > *Response to Questions*
> >
> > Some questions are addressed above.
> >
> > **Frozen grader**
> >
> > The RL baseline in Figure 6(b) and Table 4 is training with a frozen grader (Qwen3-32B) and task-specific rubrics described in the paper. Training with self-rewarding further **enhance the final performance** while **using less than 50% GPU hours**.
> >
> > We have indeed observed such behavior in the paper. As shown in Figure 7, the trained model exhibits a clear increase in Completeness, while Communication Quality (which includes readability, conciseness, etc.) decreases slightly. This pattern arises because the distribution of rubrics is imbalanced: Completeness accounts for **39% of all criteria**, whereas Communication Quality accounts for only **6%**.
> >
> > To optimize the overall composite score, the model tends to produce **longer and more exhaustive responses**, which improves completeness but can harm readability. In the appendix, we also provide qualitative examples of both performance gains and performance drops after training. Some of the failure cases are directly related to this behavior—such as the model producing an excessively long table that becomes difficult to read, resulting in a penalty.
> >
> > Thank you for your careful suggestions. We have fixed the issues in the references—one was caused by a bug in arXiv’s BibTeX export, and the other came from a Zotero plugin that inserted a note section into the metadata. The additional experimental results will be included in the revised version once we finish organizing the data. Thank you again.

---

> > ### Comment · Reviewer_GKXq · 2025-11-21
> >
> > Thanks for these new experiments. Even though I do really acknowledge your efforts in the new experiments, I would suggest it will be more convincing to show how good your method is on 3 or more different datasets/benchmarks. I would increase my score to 6 to acknowledge your efforts in solving my concerns and conducting new experiments. However, I still think my major concerns are not well addressed.
> >
> > While you do mention  "some of the failure cases", I am still confusing about do you treat it as a reward hacking? If so, how do you solve that?

---

> > > ### Author Response · Authors · 2025-11-27
> > >
> > > **More benchmarks**
> > >
> > > Thank you for quick response! We test our method on EQ-Bench 3 (measuring emotional intelligence) and IFEval (instruction following) as well. Both are tested on Qwen3-8B, where 200/1000 synthesis samples are used. The results are shown below.
> > >
> > > For EQ-Bench 3 (judged by Kimi-K2-Instruct, rubric score),
> > >
> > > - Qwen3-8B: 61.55
> > > - Qwen3-8B SR, step 60: 66.20
> > > - Qwen3-8B SR, step 120: **69.05**
> > >
> > > For IFEval, which Qwen3-8B is pretty good at, we increase samples per prompt to 16 and run for 3 epochs, the results (prompt strict) are evaluated using lm-evaluation-harness.
> > >
> > > - Qwen3-8B: 0.7726±0.0180
> > > - Qwen3-8B SR, step 10: 0.7985±0.0173
> > > - Qwen3-8B SR, step 20: 0.8115±0.0168
> > > - Qwen3-8B SR, step 30: **0.8133±0.0168**
> > >
> > > The model reaches saturation by the end of the first epoch. Another interesting observation is that we have to lower grading temperature to 0.2 for a more stable grading behavior for Qwen3-8B, otherwise it would produce ultra long reasoning trace which will influence score extraction and training speed.
> > >
> > > But we have to admit that currently there's no other benchmark as challenging and comprehensive as HealthBench. With 48,562 unique rubric criteria, the training/evaluation is more stable and convincing.
> > >
> > > **Reward hacking**
> > >
> > > Sorry about my word choice and the lack of context in my previous response. By failure cases, I mean situations where the model prioritizes producing more accurate and complete responses while ignoring readability requirements. We do not consider this reward hacking; rather, the model is learning the implicit preferences encoded in the benchmark. In fact, when the criteria are highly detailed and comprehensive, reward hacking becomes much more difficult. Of course, to correct this behavior, we would need to more carefully reweight the different scoring dimensions during later stages of training—for example, by increasing the weight of the readability score.

---

### Official Review · Reviewer_kgBb · 2025-11-01

**Soundness:** 3
**Presentation:** 3
**Contribution:** 4
**Rating:** 4
**Confidence:** 3

**Summary:**

This paper proposes a self-reward reinforcement learning method based on rubrics for handling open-ended reasoning tasks. The model evaluates its own outputs using rubrics (primarily derived from the HealthBench dataset) and employs these self-generated scores as rewards during GRPO training, while removing the KL divergence term and relying on scoring criteria for normalization. The system trained the Qwen3-32B model on the HealthBench-Easy dataset. When evaluated against GPT-4.1, it outperformed both GPT-5 and o3 models on the HealthBench-Hard dataset. The paper also examines training efficiency, dataset effects, and the SFT collapse phenomenon.

**Strengths:**

This paper highlights the challenges of open-ended evaluation in real-world reasoning, arguing that task-specific scoring criteria can provide models with stable and interpretable reward signals.

On the HealthBench Hard test, the self-reward-based GPT-4.1 model achieved a score of 0.500, outperforming GPT-5 (0.462) and o3 (0.32). This result holds consistently across two scoring temperatures (0.6 and 1.0).

Compared to baseline reinforcement learning approaches, this method reduces reward computation time by half, decreases step time by approximately 25-35%, and requires fewer GPUs.

Beyond reporting final scores, this paper provides detailed analysis of post-training changes across evaluation dimensions: for instance, improvements in completeness and context awareness, though communication quality occasionally declines with longer responses. The study also quantifies the frequency of positive and negative shifts across dimensions within the sample, exploring trade-offs rather than merely presenting isolated benchmark values.

**Weaknesses:**

All training and evaluation are confined to the HealthBench medical benchmark dataset. This limits the model's extrapolation capabilities to open domains such as law, finance, or writing.

The reward evaluator belongs to the same model family as the policy model, utilizing only the previous checkpoint and removing the KL divergence penalty term. This may introduce risks of reward drift or self-confirmation bias. The paper primarily employs score normalization and truncation based on scoring criteria but does not propose stronger safety controls or human verification mechanisms.

Core experimental results use GPT-4.1 as the final evaluator instead of human experts. The study also indicates evaluator disagreement, with some weaker evaluators tending toward leniency, raising concerns about score stability among evaluators.

Under hybrid objective settings incorporating scoring data, the Qwen3-8B model exhibited instability. The paper notes that Qwen3-8B “crashed after approximately 600 training steps due to repetitive outputs,” while Qwen3-32B remained stable, without proposing any mitigation strategies.

**Questions:**

When evaluators share the same strategy and no KL constraint is applied, how is the stability of reward behavior ensured? Can quantitative evidence of reward drift or bias be provided?

Has GPT-4.1's scoring of 0.500 results been validated through human assessment? If not validated, what is your confidence level in GPT-4.1's consistency with expert judgment?

Can the proposed method be extended to open benchmarks outside the medical domain?

Can you explain how setting the scoring temperature to the expansion temperature improves the final score? Is it because it reduces the mismatch between how answers are generated and how they are scored?

Regarding the collapse phenomenon in 8B models, what mitigation strategies (e.g., entropy regularization, output diversity penalties) were tested?

---

> ### Author Response · Authors · 2025-11-19
>
> We would like to express our sincere gratitude for your thoughtful feedback and valuable suggestions.
> Since the weaknesses and questions you raised overlap, we will address them together.
>
> **Evaluation outside medical domain**
>
> We chose HealthBench Hard because it provides a rubric-based evaluation setting that is sufficiently challenging (even frontier models score below 50% accuracy). As a complement, we also conducted experiments on the Creative Writing V3 task from EQ-Bench, which measures a model’s creative-writing capability.
>
> Specifically, we used 1,000 synthetic creative-writing prompts and applied 22 rubrics (e.g., Emotionally Engaging, Weak Dialogue) that allow the model to self-evaluate during training. The scoring strategy matches the one used in the paper: the model evaluates one rubric at a time and aggregates the scores at the end.
>
> We trained Qwen3-8B with a configuration similar to the main experiments for 10 epochs (310 steps), and used claude-sonnet-4-20250514 for grading. The results are shown below:
>
> | model        | grading_temp | step | score |
> |--------------|--------------|------|-------|
> | Qwen3-8B     | -            |  -   | 51.83 |
> | Qwen3-8B-SR  | 0.6          | 155  | 61.82 |
> | Qwen3-8B-SR  | 0.6          | 310  | 61.05 |
> | Qwen3-8B-SR  | 1.0          | 155  | 63.22 |
> | Qwen3-8B-SR  | 1.0          | 310  | 59.40 |
>
> As shown, we observe consistent improvements across different grading temperatures, suggesting that our method generalizes outside the medical domain.
>
> **Reliability of GPT-4.1 as judge**
>
> According to the HealthBench paper, GPT-4.1 achieves **physician-level agreement**. The authors designed HealthBench Meta to measure the evaluation capabilities of models or physicians, where GPT-4.1 surpassed the average performance of physicians across 5 of 7 themes. Therefore, in our paper, we first conduct more frequent evaluations using Qwen3-32B or Kimi-K2-Instruct (as shown in Figure 4 or Figure 6(b)), and finally use GPT-4.1 for scoring. We find that the results are highly **consistent across evaluators**. Regarding comparison with human experts, our perspective aligns with that of the HealthBench paper: state-of-the-art large language models have already achieved evaluation capabilities approaching or exceeding those of physicians, though human experts would be a better choice but it's too expensive. Further in-depth discussion of this topic (LLM-as-a-judge) is beyond the scope of this paper, and we leave it for future work.
>
> **Fix for 8B collapsion**
>
> Thank you for the keen observation! We carefully investigated the cause of the mid-training crashes in Qwen3-8B and found that the primary issue was that the model frequently failed to generate valid scores during evaluation due to repetitive response, causing scores to fall back to default values. By setting `frequency_penalty=0.2`, we were able to resolve this problem. The model can now train stably to completion under this configuration, and the HealthBench Hard score improved from 0.108 to 0.363. This aligns with our intuition that stable reward signals are beneficial for RL training.
>
> **Discussion on KL**
>
> The model’s scoring ability on HealthBench is reflected through its HealthBench Meta score, which measures how well the model’s rubric-based judgments **align with physicians' judgement**. As shown by the green curve in Figure 6(a), we monitor the Meta score throughout training and observe that it improves steadily. This stability is one of the main reasons we are confident in the reliability of the method.
>
> In addition, the evaluation is based on task-specific rubrics, which decompose the reward into relatively simple subtasks—for example, checking whether a response exceeds 500 words or whether it advises the patient to seek urgent care. Because these subtasks are straightforward and locally verifiable, concerns about reward drift or bias in the LLM-as-a-judge setup are less applicable.
>
> That said, the benchmark’s rubric design itself may carry biases. For instance, certain rubrics reward completeness more heavily, which can incentivize the model to generate longer responses. This explains why training tends to shift the model toward producing more verbose outputs, rather than being evidence of reward drift.
>
> Furthermore, we conducted the same experiment on Qwen3-8B with KL penalty. The final HealthBench Hard score is 0.289, which is lower than the w/o KL run (0.363), aligning with the DAPO paper.

---

> > ### Author Response · Authors · 2025-11-19
> > **(Continued)**
> >
> > **Tunable grading temperature**
> >
> > This is an excellent question. In our experiments, we used two grading temperatures: the default temperature recommended for Qwen3, and 1.0, which matches the rollout sampling temperature used during training. Both settings outperform the baseline, but the 1.0 configuration eventually achieves a higher final score.
> >
> > Our intuition is that **more accurate and informative scores** lead to more stable and effective training. This is supported by the creative-writing experiments discussed earlier. EQ-Bench provides **JudgeMark**, which evaluates a model’s ability to judge the creativity of text (analogous to HealthBench Meta). The average JudgeMark scores across three runs for each checkpoint are:
> >
> > - Original Qwen3-8B:
> >   - 0.273 ± 0.035 (grading temp = 0.6)
> >   - 0.282 ± 0.011 (grading temp = 1.0)
> > - Qwen3-8B-SR (grading temp = 0.6):
> >   - 0.307 ± 0.012 (step = 155)
> >   - 0.285 ± 0.017 (step = 310)
> > - Qwen3-8B-SR (grading temp = 1.0):
> >   - 0.268 ± 0.030 (step = 155)
> >   - 0.240 ± 0.044 (step = 310)
> >
> > We also observe that training with a grading temperature of 0.6 is more stable overall. Although the 1.0 setting performs slightly better at step 155 on Creative Writing V3, its performance drops more noticeably when overtrained, indicating reduced stability.
> >
> > Analyzing the effect of grading temperature on HealthBench is less straightforward. In Figure 7, we add the per-topic scores of the model trained with grading temp = 0.6, and compare them with the model trained at 1.0. We find that the 1.0 model performs notably better on Context Awareness, while the 0.6 model performs better on Instruction Following. Since Context Awareness accounts for **20.6%** of all rubrics, whereas Instruction Following accounts for only **4.6%**, the 1.0 configuration gains an advantage in the overall score.
> >
> > We further compared their HealthBench Meta scores and found them nearly identical. However, within the evaluator's sub-dimensions, the 1.0 model scores higher on `cluster:context_seeking`, which is likely responsible for the final performance differences.
> >
> > In summary, the effect of grading temperature is a trade-off:
> >
> > - Higher temperatures introduce **more exploration** and may lead to better final results, but can make training less stable.
> > - Lower temperatures produce more deterministic and **stable scoring**, but may limit the potential upper bound of performance.
> >
> > Thank you very much for your detailed and insightful feedback. Your comments were extremely valuable in helping us refine the work.

---

### Official Review · Reviewer_otVK · 2025-11-01

**Soundness:** 2
**Presentation:** 2
**Contribution:** 2
**Rating:** 6
**Confidence:** 3

**Summary:**

The paper introduces Self-Rewarding Rubric-Based Reinforcement Learning (SR-RBRL) for open-ended reasoning. Instead of relying on a separate reward model, the policy model grades its own outputs using task-specific rubrics, and those composite rubric scores drive RL updates.
The method omits the KL penalty used in standard GRPO (similar to DAPO), aiming to keep rewards verifiable (via explicit rubrics) while cutting scoring/compute overhead.

On HealthBench (a dialogue-style, rubric-graded medical benchmark), SR-RBRL improves performance and training efficiency: e.g., Qwen3-32B trained only on the Easy subset surpasses GPT-5 (and o3) on HealthBench-Hard, and step time/reward time drop by 25-52% under the authors’ setup. Also, The authors show that grading ability is preserved (slightly improved) after RL, and that adding a small amount of teacher-graded data helps weaker models (e.g., 8B) more than stronger ones (32B).

**Strengths:**

1. I think overall the task-specific rubric criteria and the model itself as grader is a neat design, avoiding a separate reward model and aligning rewards with what is actually evaluated.
2. With only the Easy split for training, Qwen3-32B exceeds GPT-5/o3 on HealthBench-Hard, supporting the central claim.
3. Beyond performance, Post-RL models show slight Meta-score gains as grading ability is maintained.

**Weaknesses:**

1. The paper shows response length rising with training alongside reward/score improvements, but the readability/conciseness impact isn’t systematically measured. It will be interesting to see if the model behavior or other generation pattern switches during the training.

**Questions:**

1. Results are concentrated on HealthBench (medical); it remains unclear how robust the approach is across other open-ended domains (education, legal, multi-domain assistants). Is there any testing scenario already? If so, do you think SR-RBRL will also fit?

---

> ### Author Response · Authors · 2025-11-19
>
> Thank you for your careful reading and constructive comments.
>
> **Response to Weakness 1**
>
> Regarding readability and conciseness, these aspects are generally **negatively correlated with response length**. The benchmark already includes rubrics that evaluate this behavior, such as a negative score for "Response is longer than 500 words". In Figure 6(a), we show how response length evolves during training: the model gradually increases its output from around 1k tokens to nearly 6k tokens (the configured upper limit).
>
> Interestingly, Figure 6(b) indicates that the model’s overall score continues to improve even after the response length saturates. This is further reflected in Figure 7, where the Communication Quality dimension (approximately **6% of all rubrics** and closely related to readability/conciseness) decreases slightly, while Completeness (**39%**) increases significantly. This behavior aligns with the design of the method: the model optimizes for the composite overall score, not for individual rubric categories.
>
> Additionally, the appendix provides two sets of examples, each accompanied by the corresponding changes in evaluation metrics. For instance, in the *Performance Drop Case*, the trained model receives a penalty for triggering the rubric "Uses a long table that requires horizontal scrolling on narrow viewports". This example highlights that the rubrics encourage the model to generate longer and more comprehensive responses, improving overall scores, but at the cost of slightly reduced readability and conciseness.
>
> **Response to Question 1**
>
> We chose HealthBench Hard because it provides a rubric-based evaluation setting that is sufficiently challenging (even frontier models score below 50% accuracy). As a complement, we also conducted experiments on the Creative Writing V3 task from EQ-Bench, which measures a model’s creative-writing capability.
>
> Specifically, we used 1,000 synthetic creative-writing prompts and applied 22 rubrics (e.g., Emotionally Engaging, Weak Dialogue) that allow the model to self-evaluate during training. The scoring strategy matches the one used in the paper: the model evaluates **one rubric at a time** and aggregates the scores at the end.
>
> We trained Qwen3-8B with a configuration similar to the main experiments for 10 epochs (310 steps), and used claude-sonnet-4-20250514 for grading. The results are shown below:
>
> | model        | grading_temp | step | score |
> |--------------|--------------|------|-------|
> | Qwen3-8B     | -            |  -   | 51.83 |
> | Qwen3-8B-SR  | 0.6          | 155  | 61.82 |
> | Qwen3-8B-SR  | 0.6          | 310  | 61.05 |
> | Qwen3-8B-SR  | 1.0          | 155  | 63.22 |
> | Qwen3-8B-SR  | 1.0          | 310  | 59.40 |
>
> As shown, we observe consistent improvements across different grading temperatures, suggesting that our method generalizes beyond the medical domain.
>
> We hope our responses address your questions, and we thank you again for your helpful input.

---

### Author Response · Authors · 2025-11-19

Dear reviewers, the responses for each of you have been uploaded. For convenience, some reponses have been repeated in each individual comment section. If you spot any typos or need further clarification on any point, please let me know. Thank you.

---

### Meta-Review · Area_Chair_WeY3 · 2025-12-23

**Summary:**

The authors propose a self-reward reinforcement learning method based on rubrics. Most reviewers' initial scores hovered near the borderline, with one reviewer recommending rejection. The authors submitted a rebuttal, and one reviewer engaged in the discussion. After thoroughly reviewing the paper and the authors' responses, the AC concluded that reviewers' concerns regarding the reward cheating mechanism, methodological innovation, and reliability remained unresolved. The authors are advised to revise these aspects and submit to a subsequent conference.

**Reviewer Concerns:**

Concerns related to HealthBench have been resolved, but I believe concerns raised by other reviewers remain inadequately addressed.

**Reviewer Scores:**

I believe the reviewers will maintain their scores.

---

### Decision · Program_Chairs · 2026-01-26

Reject